# Cost of Care and Pattern of Medical Care Use in the Last Year of Life among Long-Term Care Insurance Beneficiaries in South Korea: Using National Claims Data

**DOI:** 10.3390/ijerph17239078

**Published:** 2020-12-04

**Authors:** Sunjoo Boo, Jungah Lee, Hyunjin Oh

**Affiliations:** 1Research Institute of Nursing Science, College of Nursing, Ajou University, Suwon 16499, Korea; 2Laboratory of Emergency Medical Services, Seoul National University Hospital Biomedical Research Institute, 71, Daehak-ro, Jongno-gu, Seoul 03082, Korea; jalee9406@gmail.com; 3College of Nursing, Gachon University, Incheon 98105, Korea

**Keywords:** aged, health care costs, insurance, long-term care

## Abstract

In Korea, a substantial proportion of long-term care insurance (LTCI) beneficiaries die within 1 year of seeking the benefit. This study was conducted to evaluate the pattern of medical care use and care cost during the last year of life among Korean LTCI beneficiaries between 2009 and 2013 using the national claims data. The National Health Insurance’s Senior (NHIS-Senior) cohort was used for this retrospective study. The participants were LTCI beneficiaries aged 65 or over as of 2008 who died between 2009 and 2013 (*N* = 30,433). Medical costs during the last year of life were highest for those who used both medical care services and long-term care (LTC) services and increased as death approached. About half of the participants were hospitalized at the time of death. The use of LTC services at the time of death increased from 13.0 to 22.8%, while those who died at home decreased from 34 to 20%. This study suggests that the use of LTC services did not reduce medical costs by substituting unnecessary inpatient hospitalization. Quality of dying should be considered one of the goals of older adult care, and provisions should be made for palliative care at home or LTC facilities.

## 1. Introduction

With increasing life expectancy and a declining birth rate, Korea is the most rapidly aging country in the world. The proportion of those aged 65 or over was 7% in 2000, 14% in 2017, and is expected to reach 20% in 2025 [1]. The life expectancy of Korean men and women in 2030 is estimated to reach 84.1 and 90.8 years, respectively, which is the longest among the 35 industrialized countries [2]. Such a rapidly aging population leads to a substantial socioeconomic burden. Approximately 41% of total medical expenses in Korea are attributable to those over 65 years old [3], and this rate is expected to increase further.

As a result of rapid aging, a substantial share of resources is being devoted to dying people. Between 2005 and 2015, the cost of care in the last year of life increased approximately 3.5-fold among Koreans aged over 65 years. In addition, expenses tended to increase as death approached, and 41.8% of the total medical cost during the last year of life was used in the last 3 months [4]. High cost during the last 3 months of life suggests that many Korean older adults die in hospital with “lifesaving” treatments, although most prefer to die at home without meaningless and ineffective treatments [5].

South Korea has a system called the National Health Insurance Service (NHIS) in the form of social security. About 97% of the population is covered by Health Insurance, a compulsory social insurance program, and the remaining population is covered by Medical Aid, a program to help with medical costs for people with limited income and resources. Given that the Korean society is aging more rapidly than any other country, an additional social insurance—Long-Term Care Insurance (LTCI)—was introduced in 2008. This provision is for those aged 65 years or over who need care/support for daily living or persons under 65 years who have geriatric diseases, so as to prevent or slow down the decline of functionality via home and community-based services as well as to reduce medical costs. In 2018, 8.4% of Korean older adults benefited from LTCI [6].

Those who are eligible for LTCI benefits can choose between institutional care and home care [7]. LTCI services cover various home care and institutional care services. Home care includes home visiting nursing, home visiting care, home bathing, day and night care, and short-term respite care in the community. Institutional care under the LTC services includes long-term nursing care and rehabilitation services in an institution [8]. About 20% of LTCI beneficiaries die within 1 year of utilizing the service [9]. This suggests that of all deaths of those aged over 65, the proportion of deaths of LTCI beneficiaries may be substantial. Therefore, how and where LTCI beneficiaries spend their end of life (EOL) has a great impact on their quality of life and the medical cost for the patient/family and society. Given the substantial number of deaths among LTCI beneficiaries and the high medical cost during the last phase of life, it is important to evaluate LTCI beneficiaries’ cost and use of care in the last year of life, which could be an indirect measure of the quality of dying. A few studies have reported on the cost and use of care in the last year of life among patients with cancer in Korea [10], but no study has focused on LTCI beneficiaries.

Perspectives on the effects of LTCI on the cost of care and medical utilization are inconsistent. A study by Forder [11] suggests that LTCI in older adults decreases medical costs by reducing meaningless hospitalization and hospital stays especially at the end of life. However, Noh [12] suggests that LTC services are not likely to substitute medical needs of older adults leading to an increase in the health care expenditure. There exists an opportunity to examine whether LTCI reduces medical costs by decreasing hospitalization especially at the end of life. Notably, several patterns of care used at the end of life are examined in a previous study [13]. In the study, hospice services and medical costs received during the last 3, 10, and 30 days of life were analyzed. Therefore, in order to examine whether LTCI reduces medical costs, the aim of this study was to evaluate the pattern of use of medical care and their cost during the last year of life among LTCI beneficiaries in Korea between 2009 and 2013, using national administrative claims data. Additionally, we examined the place of death, number of hospital admissions, and average inpatient days at 90, 30, 10, and 3 days before death. This was based on earlier studies that have shown that one fifth of older people in LTC facilities in Korea were hospitalized at least once during the last 90 days [14] and about one third of LTCI beneficiaries received life-sustaining treatments during the last 30 days of life [15]. The results of this study could provide the necessary context to inform discussions of new models of EOL care for this population.

## 2. Materials and Methods

### 2.1. Data Source and Study Design

The National Health Insurance Service-Senior (NHIS-Senior) cohort was used for this retrospective cohort study [16]. As the NHIS covers the entire population of Korea as a single public insurer, all records of healthcare and LTC services are stored in their database as national administration data. The NHIS-Senior includes information from elderly individuals randomly sampled by the NHIS database. The cohort consists of five databases: an eligibility database, a national health screening database, a healthcare utilization database, a LTCI database, and a healthcare provider database [16]. The NHIS-Senior cohort was established to provide public health researchers and policy makers with representative information regarding older adults’ utilization of medical and LTC services. The original cohort of about 550,000 older adults aged over 60 years was selected in 2002 and followed up for 11 years until 2013, unless they were disqualified owing to death or emigration [16]. Since the NHIS database is the national administrative claims database, NHIS-Senior is nationally representative with very low missing data or attrition rate. The cohort datasets include de-identified information to protect patient anonymity, and alternative identification numbers are assigned to facilitate matching descriptive data on the same patient within the entire dataset.

Our sample was restricted to LTCI beneficiaries who were aged 65 or over as of 2008 and died between 2009 and 2013 (*n* = 33,224). Among them, those who died before July 2009 were excluded (*n* = 2791) because LTCI was introduced in July 2008 and this study evaluated the medical cost of LTCI beneficiaries for 1 year before death; thus, the final sample for analysis was 30,433. The study’s retrospective protocol was approved by the appropriate institutional review board (AJIRB-SBR-EXP-17-467).

### 2.2. Measures

The NHIS-Senior includes information on date of death, socio-economic characteristics, medical care utilization in hospitals, and types of hospitals (clinic, general hospital, and tertiary hospitals). It also includes information regarding LTCI, such as grade, service utilization, and related cost. The LTCI grade is categorized based on the level of activities of daily living: Grade 1 (critical needs), Grade 2 (substantial needs), and Grade 3 (moderate needs) [17]. Based on the individual’s medical care utilization 1 year before death, patterns of healthcare utilization were categorized into four types in this study as follows: (1) only medical care (MC) user during the last year, (2) both medical and LTC service user (MC + LTC), (3) only LTC service user, and (4) user of neither.

Medical expenses incurred in medical care settings (inpatient, outpatient, and drug prescription) or LTC settings (LTC facilities or in-home services) were calculated. The cost was a combination of both insurer and out-of-pocket payments for each claim. Additionally, the number of inpatient days was calculated for Group 1 and 2 within the last 90 days of life. The place of death was classified as “died in hospital” if there were claim data for use by the medical institution on the day of death. Likewise, the place of death was classified as “died in LTC facility” if the claim data for LTC facilities were available on the day of death. Those with no claim data on the day of death were classified as “died at home.”

### 2.3. Statistical Analysis

All analyses were performed using SAS software ver. 9.4 (SAS Institute Inc., Cary, NC, USA). Descriptive analyses were performed to present participants’ characteristics, pattern of medical care use, place of death, and healthcare expenditures 1 year before death. The ANOVA test was used to evaluate the differences in the average healthcare expenditure 1 year before death by pattern of medical care use. Chi-squared tests were used to evaluate differences in the characteristics of participants by pattern of medical care use. Two-sided *p*-values less than 0.05 were considered as indicating statistical significance.

## 3. Results

The subjects included 30,433 LTCI beneficiaries who died between 1 July 2009 and 31 December 2013. About 63% were women and about 68% were 80 years old or older. Primary care givers were their adult children. About 29% died due to cardiovascular disease, followed by neoplasms (14.59%). The general characteristics and cause of death of subjects in this study are similar to the study by Han et al. [15], which reviewed all LTCI decedents’ data from 2008 to 2012 in NHIS. The pattern of healthcare utilization is summarized in Table 1. About three in four subjects (*n* = 23,162) received both MC services and LTC services during the last year of life; 22% (*n* = 6713) received only MC services covered by NHIS. The pattern of medical care use differed by participant characteristics. Gender (female), older age, higher LTCI grade, lower household income, and children as primary caregivers were associated with receiving LTC services (Table 2).

Medical cost during the last year of death by pattern of healthcare utilization is depicted in Figure 1. The average healthcare expenditure one year before death was statistically significantly different by pattern of medical care use (F = 209.81, *p* < 0.001). Our results imply that the introduction of LTC has led to the use of MC + LTC by the majority of people and has not lowered the cost compared to only MC. During the last year of life, Group 2 (MC + LTC) incurred the highest medical care cost of USD 18,717, followed by Group 1 (MC) with USD 17,721, and Group 3 (LTC) with USD 7292. Medical care cost for Groups 1 and 2 generally increased as death approached and peaked 1 month before death. The rate of increase for Group 1 showed a very steep rise 3 months before death. The amount of money that Group 1 spent in the month before death was approximately three times that of Group 3.

Hospitalizations and hospital stay for Groups 1 and 2 were examined in the last 90 days of life (Table 3). About 84% of Group 1 were admitted to hospital 30 days before death at least once, while the rate was 62% for Group 2. Number of hospital days for those admitted to hospital 30 days before death was 26 days for Group 1 and 19 days for Group 2. Table 4 shows the place of death of participants. About half of decedents from 2009 to 2013 were hospitalized at the time of death. Among all decedents, the use of LTC services at the time of death increased from 13.0% in 2009 to 22.8% in 2013, while those who died at home decreased from 34% in 2009 to 20% in 2013.

## 4. Discussion

Korea is the most rapidly aging country in the world. Medical costs for those older than 65 years were estimated to be about 40% of all medical costs in Korea in 2019 [3]. A large share of overall lifetime medical cost, about 20–25%, occurs in the last year of life [18,19]. Given the concerns over the high cost and often minimal health benefits during the last months of life [10,11,15,19], discussions on the quality of dying should be considered as one of the goals of older adult care. How and where to spend the last year of life has a great impact on quality of life and medical cost [20,21]. We analyzed the pattern of healthcare use, place of death, and medical cost of LTCI beneficiaries in the last year of life to provide the necessary context for discussions of quality of older adult care.

Between 2009 and 2013, the annual number of deaths increased from 2896 to 7229 as per this study. This increase was commensurate with an increase in LTCI beneficiaries, from 2.9% of older adults in 2008 to 5.9% in 2013 [6]. In a recent study using public administrative data, about three-quarters of older adults with dementia and cancer died in hospital [22]. The rate of Korean older adults dying in hospitals is high compared with other countries [13,23]. Older adults are more likely to experience delirium, sustain fall injuries, and undergo unnecessary procedures in the hospital environment [24]. Whether these procedures and treatments have a positive effect on the quality of life of patients nearing EOL needs careful evaluation. These procedures are the main cause of the high cost of care in the last months prior to death.

A survey of Korean older adults indicated that home is the preferred place of death [22,25]. However, the proportion of participants in this study who died at home declined during the study period (from 34 to 20%), whereas the proportion of those who died in LTC facilities increased (from 13 to 26%); the remaining participants (about half) died in hospitals, and the rate remained relatively stable. This trend is different from a study with Medicare beneficiaries in the United States [19] that showed an increase in deaths at home (from 30.7% in 2000 to 40.1% in 2015) and a decrease in deaths in acute care hospitals (from 32.6% in 2000 to 19.8% in 2015). The authors suggest that such changes in the site of death in the United States are attributable to the large increase in the use of hospice and palliative services, which reduce inpatient hospitalization. What proportion of hospital deaths would be appropriate is indeed a hard question, but the findings from this study suggest that the introduction of LTCI in Korea did not affect the rate of deaths that occurred in the hospital. As the preference of family members regarding the place of death might strongly impact patients’ preferred place of death and actual place at the EOL [26], the increasing rates of death in LTC facilities indicate recent trends after the introduction of LTCI in Korea.

The public LTCI system was introduced in 2008 in Korea to prevent the decline of physical function among older adults as well as to reduce familial care burden through the use of community-based services. It was expected that medical cost would be reduced by substituting meaningless inpatient hospitalization with LTC services. However, we did not observe the expected lower use of medical services. The majority of participants (76.1%) used LTC services covered by LTCI along with medical services covered by NHIS (Group 2). About 22% used only medical care covered by NHIS. Only 1.3% of participants used only LTC services in the last year of life (Group 3). For Group 3, the average medical cost during the last 1 year was less than half of the others. Transition to acute care hospitals and/or medical treatment covered by NHIS increases medical costs. This means that LTC services did not substitute medical needs of a majority of LTCI beneficiaries, suggesting that population ageing in Korea would further increase the socio-economic burden for dying people unless appropriate EOL services are introduced in LTC facilities.

In this study, the total average cost during the last year of life was highest among those who received both services covered by NHIS and LTCI (USD 18,717), followed by those who received medical services only (USD 17,721). Interestingly, the average medical cost of Group 2 was higher than that of Group 1 from 12 months before death to 4 months before death. During the last three months of life, the medical cost for Group 1 sharply increased and was higher than that for Group 2. Even though further investigation of why 22% of LTC beneficiaries did not use LTC services at all during their last year of life is needed, the high cost of the last three months of life may be partly due to inpatient hospitalization and/or intensive treatment. About 84% of Group 1 and 62% of Group 2 participants were hospitalized at least once before the last 30 days of life. Average hospital days for Group 1 and 2 during the last 30 days of life were 25.59 and 18.98 days, respectively. The rate of hospitalization decreased as death approached, but 78% of Group 1 and 55% of Group 2 participants were hospitalized during the last three days of life. Such transitions, such as transfer from an LTC facility to acute care hospitals seem to be ineffective. Especially during the last few days of life, such hospitalizations do not adequately address the older adults’ special needs and lack of communication regarding goals of care [27]. However, quality of dying should be considered one of the goals of care for LTCI beneficiaries. The high rates of inpatient hospitalization are also in part because LTC facilities are not healthcare organizations and are primarily designed to provide support for daily living rather than focusing on medical care needs. About 90% of the workforce in LTC facilities in South Korea were care assistants, 0.7% were registered nurses (RNs), and 2.5% were nurse aids in 2018 [6]. No RNs are available in about 80% of LTC facilities in Korea [28], which is different from nursing homes in Western countries. However, all residents in LTC facilities are known to have at least one chronic disease, and 54% have two or more, thus requiring medical care [29]. LTC facility residents are supposed to be transferred to hospitals whenever they need acute medical care. In this study, rates of hospitalization were high during the last 30 days of life. Such a terminally-ill-stage transfer would be ineffective, inefficient, and could raise safety issues.

Past literature suggests hospice care to be cost-effective, as it prevents costly hospitalization [30,31]. In South Korea, hospice and palliative care are not included in LTC services. Hospice and palliative care are meant to improve the quality of life of those with life-threatening diseases and their families by decreasing their physical, psychological, social, and spiritual distress [32]. About 48% of LTCI beneficiaries were in LTC facilities or at home when they died, implying an increasing trend in the number of older adults dying in LTC facilities. Long-term exacerbation issues among older adults relate to dependence, social isolation, and family burden, such that they would not return to their homes once they are in LTC facilities. This implies that EOL care at home and LTC facilities are necessary [25]. The perception of EOL care among older adults and their families should also be changed regarding the determination of the place of death and be supported by public education.

There are several issues that may have affected the accuracy of the findings of this study. This study analyzed the NHIS-Senior cohort. The cohort is based on national administrative claims databases and thus includes valid and accurate information especially for socio-economic status, healthcare utilization, and death information with a very low missing or attrition rate. NHIS-Senior is nationally representative of the entire elderly population but not especially of LTCI beneficiaries. However, the general characteristics and cause of death of subjects in this study are similar to the study by Han et al. [15], which reviewed all LTCI decedents’ data from 2008 to 2012 in NHIS. No information of place of death is available in the original dataset; thus, we classified those with no claim data on the day of death as “died at home.” This classification ignores those who might have died while being transported to receive treatment in the hospital or those with missing data. In addition, this study analyzed the pattern of medical care use and cost of care in the last year of life among LTCI beneficiaries from 2009 to 2013 because it was the latest dataset available for researchers, but the pattern might change over time. These limitations should be considered in order to appropriately interpret the results of this study.

## 5. Conclusions

This study elucidated that the proportion of older adults who died at home declined and those who died in LTC facilities increased between 2009 and 2013. The use of LTC services did not reduce the medical cost by replacing meaningless inpatient hospitalization. Older adults hope to die at home; therefore, provisions should be made for palliative care at home or LTC facilities.

## Figures and Tables

**Figure 1 ijerph-17-09078-f001:**
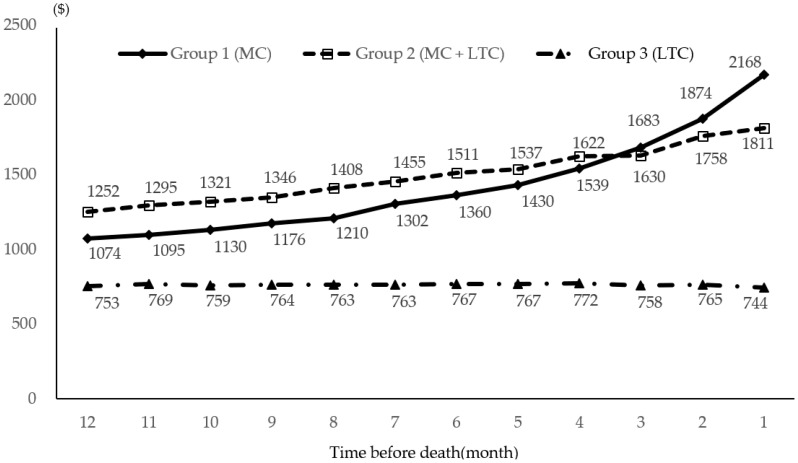
Average medical costs during the last year before death among long-term care beneficiaries. Note: numbers are means; USD 1 = KRW 1130.50.

**Table 1 ijerph-17-09078-t001:** Pattern of medical care use of long-term care insurance (LTCI) beneficiaries.

Types of Medical Care Use		*n*	(%)
MC		6713	(22.06)
MC + LTC	MC + LTC-home care	12,280	(40.35)
	MC + LTC-facility care	7921	(26.03)
	MC + LTC-home & facility care	2961	(9.73)
LTC	LTC-home care	345	(1.13)
	LTC-facility care	44	(0.14)
	LTC-home & facility care	12	(0.04)
Neither		157	(0.52)

MC = medical care services; LTC = long-term care services.

**Table 2 ijerph-17-09078-t002:** Characteristics of subjects by pattern of medical care use.

Characteristics	Total (*N* = 30,433)		MC (*n* = 6713)		MC + LTC (*n* = 23,162)		LTC (*n* = 401)		Neither (*n* = 157)		χ^2^	*p*
	*n* (%)											
Gender												
Men	11,305	(37.15)	2659	(39.61)	8496	(36.68)	102	(25.44)	48	(30.57)	46.05	<0.001
Women	19,128	(62.85)	4054	(60.39)	14,666	(63.32)	299	(74.56)	109	(69.43)		
Age (yr)												
65–69	454	(1.49)	112	(1.67)	340	(1.47)	--	--	2	(1.27)	298.52	<0.001
70–74	3463	(11.38)	883	(13.15)	2556	(11.04)	13	(3.24)	11	(7.01)		
75–79	5735	(18.84)	1441	(21.47)	4238	(18.3)	35	(8.73)	21	(13.38)		
80–84	7270	(23.89)	1705	(25.4)	5488	(23.69)	50	(12.47)	27	(17.20)		
85 or over	13,511	(44.40)	2572	(38.31)	10,540	(45.51)	303	(75.56)	96	(61.15)		
Grade of LTCI (Level of benefit coverage)												
1	6838	(22.47)	1760	(26.22)	4920	(21.24)	115	(28.68)	43	(27.39)	94.42	<0.001
2	8384	(27.55)	1747	(26.02)	6466	(27.92)	119	(29.68)	52	(33.12)		
3	15,211	(49.98)	3206	(47.76)	11,776	(50.84)	167	(41.65)	62	(39.49)		
Insurance type												
HI	25,607	(84.14)	5757	(85.76)	19,336	(83.48)	377	(94.01)	137	(87.26)	51.16	<0.001
Medical aid	4826	(15.86)	956	(14.24)	3826	(16.52)	24	(5.99)	20	(12.74)		
Household income level												
Low	8004	(26.30)	1643	(24.47)	6244	(26.96)	77	(19.2)	40	(25.48)	65.36	<0.001
Mid-low	7649	(25.13)	1697	(25.28)	5760	(24.87)	149	(37.16)	43	(27.39)		
Mid-high	9069	(29.80)	2045	(30.46)	6852	(29.58)	111	(27.68)	61	(38.85)		
High	5711	(18.77)	1328	(19.78)	4306	(18.59)	64	(15.96)	13	(8.28)		
Primary caregiver												
Spouse	6349	(20.86)	1343	(20.01)	4936	(21.31)	41	(10.22)	29	(18.47)	1170.29	<0.001
Children	11,214	(36.85)	2316	(34.5)	8504	(36.72)	295	(73.57)	99	(63.06)		
Care assistant	7030	(23.10)	2348	(34.98)	4638	(20.02)	25	(6.23)	19	(12.10)		
None	1034	(3.40)	157	(2.34)	874	(3.77)	3	(0.75)	--	--		
Else	4806	(15.79)	549	(8.18)	4210	(18.18)	37	(9.23)	10	(6.37)		
Place of residence at applying for LTCI												
Home	19,608	(64.43)	3781	(56.32)	15,344	(66.25)	353	(88.03)	130	(82.8)	3142.68	<0.001
LTC facility	4810	(15.81)	290	(4.32)	4481	(19.35)	31	(7.73)	8	(5.10)		
Geriatric hospital	4174	(13.72)	2180	(32.47)	1988	(8.58)	1	(0.25)	5	(3.18)		
Else	1841	(6.05)	462	(6.88)	1349	(5.82)	16	(3.99)	14	(8.92)		
Cause of death												
Circulatory ds.	8784	(28.86)	2089	(31.12)	6583	(28.42)	78	(19.41)	34	(21.66)	731.61	<0.001
Neoplasms	4440	(14.59)	1303	(19.41)	3133	(13.52)	2	(0.50)	2	(1.27)		
Respiratory ds.	3685	(12.11)	647	(9.64)	3012	(13.02)	18	(4.48)	8	(5.10)		
Endocrine, nutritional ds.	1846	(6.07)	434	(6.46)	1402	(6.05)	7	(1.74)	3	(1.91)		
Nervous system ds.	1659	(5.45)	488	(7.28)	1161	(5.01)	8	(1.99)	2	(1.27)		
Others	10,019	(32.92)	1752	(26.09)	7871	(33.98)	288	(71.64)	108	68.79)		

MC = medical care services; LTC = long-term care services; LTCI = long-term care insurance; HI = health insurance.

**Table 3 ijerph-17-09078-t003:** Number of admissions and average hospital days at 90, 30, 10, and 3 days before death.

Group	Number of Admissions, *n* (%)						*n* (%)	Hospital Days, M ± SD
	1		2		≥3			
MC Group (*N* = 6713)								
90 days before death	3187	(54.80)	1005	(17.28)	1624	(27.92)	5816 (86.64)	66.77 ± 32.55
30 days before death	4280	(75.83)	944	(16.73)	420	(7.44)	5644 (84.08)	25.59 ± 9.14
10 days before death	4849	(89.58)	465	(8.59)	99	(1.83)	5413 (80.63)	9.43 ± 2.33
3 days before death	5064	(95.91)	198	(3.75)	18	(0.34)	5280 (78.65)	2.92 ± 0.76
MC + LTC Group (*N* = 23,162)								
90 days before death	8344	(52.93)	3820	(24.23)	3599	(22.83)	15,763 (68.06)	38.14 ± 32.26
30 days before death	10,453	(72.68)	2848	(19.80)	1081	(7.52)	14,382 (62.09)	18.98 ± 11.51
10 days before death	11,637	(87.04)	1468	(10.98)	265	(1.98)	13,370 (57.72)	8.19 ± 3.39
3 days before death	12,074	(94.63)	634	(4.97)	51	(0.40)	12,759 (55.09)	2.91 ± 1.01

MC = medical care services; LTC = long-term care services; M = mean; SD = standard deviation.

**Table 4 ijerph-17-09078-t004:** Place of death of LTC beneficiaries by year (*N* = 30,433).

Place of Death	Year, *n* (%)												χ^2^	*p*
	2009		2010		2011		2012		2013		Total			
Hospital	1534	(52.97)	3244	(51.89)	3272	(47.83)	3798	(52.64)	3897	(53.91)	15,745	(51.74)	456.78	<0.001
LTC facility	377	(13.02)	1252	(20.03)	1627	(23.78)	1807	(25.05)	1881	(26.02)	6944	(22.82)		
Home	985	(34.01)	1756	(28.09)	1942	(28.39)	1610	(22.31)	1451	(20.07)	7744	(25.45)		
Total	2896	(100.0)	6252	(100.0)	6841	(100.0)	7215	(100.0)	7229	(100.0)	30,433	(100.0)		

LTC = long-term care services.

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
