# Peer review of "Cost of Care and Pattern of Medical Care Use in the Last Year of Life among Long-Term Care Insurance Beneficiaries in South Korea: Using National Claims Data"

_ijerph, 2020, doi:10.3390/ijerph17239078_

Round 1

Reviewer 1 Report

Dear Dr. Boo and Dr. Oh,

Thank you for your manuscripts on palliative care research during this pandemic. Hope everything is going well.

  1. Please add the Group 3 legend label in Figure 1.
  2. On Table 3, Hospital days M ± SD, does M represent mean or median? Could you label it? Additionally, please also label the cost of each month as the mean or median value in Figure 1.
  3. Let’s start the Newcastle-Ottawa Scale (NOS) quality assessment. This is the form for retrospective cohort study evaluation.

1) Could you add some sentences to clarify the representation of the exposed and non-exposed cohort  (after line 114)?

In lines 74 to 75, what you wrote does not indicate the representativeness of the exposed/non-exposed cohorts.

Below is an example.

“The majority of deaths (58.1%) was among those 75–94 years, and occurred in urban areas (84.7%—similar to the proportion of all Ontario urban residents).There was a consistent gradient of higher proportions of decedents residing in lower income neighborhoods (p<0.0001).

Tanuseputro, Peter, et al. "The health care cost of dying: a population-based retrospective cohort study of the last year of life in Ontario, Canada." PLoS one 10.3 (2015): e0121759.

2) You mentioned “The NHI’s older adult cohort data” ( line 70),  “The cohort datasets” ( line 74), and “The older adult cohort database” ( line 85).  Could you include the exact name of the databases?

For example,

“Data were obtained from three sources: the Northern and Yorkshire Cancer Registry and Information Service (NYCRIS), SystmOne and the Patient Pathway Manager (PPM).”

Ziegler, Lucy E., et al. "Is palliative care support associated with better quality end-of-life care indicators for patients with advanced cancer? A retrospective cohort study." BMJ open 8.1 (2018).

I am sure you have the databases since you mentioned at line 98 (the data from the medical institution) and line 99 (the data from LTC facilities).

3) In lines 100 and 101, you wrote that “Those with no claim data on the day of death were classified as

 ‘died at home.’ ”  How about the missing data? I think that is why we cannot discuss follow-up in this study. Could you write about this in the discussion when addressing study limitations?

Here is the NOS (The Newcastle-Ottawa Scale) before and after modification.

4 There is already the past 7 years after the period of your cohort study.  The study I cited before is no longer than 5 years.  Could you write something about it in the discussion part?

Author Response

The manuscript has been rechecked and appropriate changes have been made in accordance with the reviewers’ suggestions. The responses to their comments have been prepared and attached herewith.

We thank you and the reviewers for your thoughtful suggestions and insights, which have enriched the manuscript and produced a better and more balanced account of the research. We hope that the revised manuscript is now suitable for publication in your journal.

Reviewer 2 Report

Korea is the most rapidly aging  country in the world. It is an interesting topic to evaluate the pattern of medical use and care cost during the last year of life among Korean long-term care (LTC) insurance beneficiaries between 2009 and 2013. 

My comments on this paper are structured as follows: theoretical framework, descriptive statistics and results.

1) Theoretical framework. 

This paper lacks a theoretical framework and the mechanism of LIC  on the cost of care and pattern of medical use in the last year of life. Here, I donot mean to have a mathematical model. But the paper should set up testing hypotheses related to the research questions and estimating results. 

2) Descriptive statistics.

In line 65, and Table 3,  you should refer to the relevant reference. Because I do not know more details about LIC, and can not understand why you choose the 90, 30, 10 and 3 days before death. Additionally, please describe in more detail about the age, income, the numbers of household member and cause of death etc. 

3) Results.

Due to the lack of theoretical framework, the current results are weak  to explain the relationship between the LIC and the medical cost.  In the revised manuscript, the more empirical results should be offered. For example, you can further  use the two-stage model (Heckman Model), Tobit Model, two-part model. 

Author Response

(The authors gave the same response as above.)

Reviewer 3 Report

This study provides an analysis of the pattern of medical use of long-term care insurance beneficiaries and the places of death based on 11 years of cohort data on elderly subjects in Korea (n=30,433). Aging is a challenge for many countries and the results of this study provide important insights into how care of end of life can be provided from an administrative, health, and care perspective.

However, as an academic paper, there are several problems in the following respects.

Firstly, the purpose of this study is not positioned in the existing knowledge body and the novelty of this study cannot be evaluated explicitly. Based on the relevant literature, the objective and research question of this study need to be established.

In methodology, the statistical analyses used are descriptive statistics and chi-square test, which do not seem to adequately explain the characteristics of the data sufficiently in terms of statistics.

The following are detailed comments tied to the line numbers.

------------------

#13: [abstract] Explain clearly where the originality or novelty of this study lies.

#44: Is Medical Aid a proper noun?

#63: In terms of academic paper, could you also show us what discussions have been developed in other literature on this topic? And then, please position the purpose and aims of this study in the context.

#125: There is no Group 3 legend.

#133: What do the three categories under ``Number of admissions mean''?

#135: If the statistical method is only a chi-square test, the independence between variables can be discussed only. If possible, could multiple statistical methods use for supporting the findings of this study? Just one of the ideas: for example, each subjects' cumulative costs from 12 months prior to death to the date of death are compared by the groups, with using analysis of ANOVA or Kruskal-Wallis test, and so on.

#140: ``minimal health benefits during the last months of life'' <- Where did this perspective come from?

#144: With regard to quality, could you explain based on the results of this study's analysis?

#148: ``The rate of Korean older adults dying in 148 hospitals is high compared with other countries.'' <- Is this a result of a comparison with some literature?

#208: ``The use of LTC services did not reduce 207 medical cost by replacing meaningless inpatient hospitalization.'' <- Could you explain the cost perspective based on the results of this study anywhere in the discussion?

------------------

Author Response

(The authors gave the same response as above.)

Round 2

Reviewer 1 Report

Dear Drs, 

You are all set. 

Thank you so much for your modification. 

Best of luck in your future endeavors!

Author Response

Thank you for much for your comments, and those were very helpful to improve our manuscript!

Reviewer 2 Report

Thank you for your job on the revised manuscript. Although the modification is not perfect, I very much approve of your modification job in this revised manuscript. More importantly,  the topic of this article is interesting. 

It is my pleasure to comment on your article. Thanks.

Author Response

Thank you for much. 

Reviewer 3 Report

Additional literature review and analysis has been done and has been adequately revised.

Below are the comments tied to the line numbers.

-------------

70: A period is missing
145: In addition to the significant difference, there may be further explanations to support this paper's claims, such as that the introduction of LTC has led to the use of MC+LTC by the majority of people and has not lowered the cost compared to only MC.

-------------

Author Response

Dear reviewer:

Very much thank you for your suggestions.

We made changed following your comments.

1) 70: A period is missing

--> We added the period. 

2) 145: In addition to the significant difference, there may be further explanations to support this paper's claims, such as that the introduction of LTC has led to the use of MC+LTC by the majority of people and has not lowered the cost compared to only MC.

--> We added the further explanation. 

Our results imply that the introduction of LTC has led to the use of MC+LTC by the majority of subject and has not lowered the care cost during the last year of life compared to only MC.

Thank you so much.

Best,

Hyunjin Oh